# Operational Impacts of On-Demand Ride-Pooling Service Options in Birmingham, AL

Furat Salman [1] , Virginia P. Sisiopiku [1,*] , Jalal Khalil [2], Wencui Yang [1] and Da Yan [2]

1   Department of Civil, Construction, and Environmental Engineering, The University of Alabama at Birmingham, Birmingham, AL 35294, USA; furat@uab.edu (F.S.); yangw@uab.edu (W.Y.)
2   Department of Computer Science, The University of Alabama at Birmingham, Birmingham, AL 35294, USA; jalalk@uab.edu (J.K.); yanda@uab.edu (D.Y.)
*   Correspondence: vsisiopi@uab.edu

**Abstract:** Transportation Network Companies (TNCs) use online-enabled apps to provide on-demand transportation services. TNCs facilitate travelers to connect with drivers that can offer them rides for compensation using driver-owned vehicles. The ride requests can be for (a) individual or (b) shared rides. The latter, also known as ride-pooling services, accommodates requests of unrelated parties with origins and destinations along the same route who agree to share the same vehicle, usually at a discounted fare. Uber and Lyft offer ride-pooling services in select markets. Compared to individual ride requests, ride-pooling services hold better promise toward easing urban congestion by reducing the number of automobiles on the road. However, their impact on traffic operations is still not fully understood. Using Birmingham, AL as a case study, this research evaluated the impact that ride-pooling services have on traffic operations using a Multi-Agent Transport Simulation (MATSim) model of the Birmingham metro area. Scenarios were developed to simulate baseline conditions (no TNC service) and ride-pooling availability with two types of ride-pooling services, namely door-to-door (d2d) and stop-based (sB) service and three fleet sizes (200, 400, and 800 vehicles). The results indicate that when TNC vehicles are added to the network, the Vehicle Kilometers Traveled (VKT) decrease by up to 5.78% for the door-to-door (d2d) service, and up to 2.71% for stop-based (sB) services, as compared to the baseline scenario (no TNC service). The findings also suggest that an increase in the size of the ride-pooling fleet results in a rise in total ride-pooling service VKT, network-wide total VKT, and detour distance. However, increasing the size of the ride-pooling fleet also results in a decrease in the ride request rejection rates, thus benefiting the customers and decreasing the vehicle empty ratio which, in turn, benefits the TNC drivers. The results further suggest that a fleet of 200 ride-pooling vehicles can meet the current demand for service in the Birmingham region at all times, thus it is the optimal ride-pooling TNC fleet size for a medium-sized city such as Birmingham.

**Keywords:** Transportation Network Companies (TNCs); ride-pooling; Uber Pool; Lyft Line; on-demand ride-sourcing; MATSim

## 1. Introduction

A ride-pooling service is an on-demand transportation option wherein unrelated users are matched to share a ride in a single vehicle. Trips are matched according to pick-up and drop-off locations to ensure riders can travel together in the same vehicle while maintaining a reasonable travel time and delay time. In recent years, ride-pooling services have gained popularity as a ride-sourcing transportation option, as they allow more than one passenger's request to be served in one ride, thus reducing the number of vehicles on the network [1]. Transportation Network Companies (TNCs) view ride-pooling services as a way to increase ridership, reduce customer cost, and expand ridesharing options [2]. Uber and Lyft, the two most popular ride-hailing companies, offer Uber Pool and Lyft Line

ride-pooling services to their customers in many cities throughout the world. According to Lo and Morseman [3], the Uber company launched the Uber Pool service in 2014 to make it easier for riders to share their trip with other travelers who are traveling in the same direction [3]. Drivers also benefit by reducing their operating costs while maximizing their revenue from shared rides. As part of an Uber Pool trip, Uber determines the best route along which multiple riders are going to be picked up [4]. In addition to offering rides at a reduced cost to customers, ride-pooling is viewed by many as a great way to reduce urban traffic congestion [1,2,5]. However, there is still a lack of clarity regarding the true impact of ride-pooling services on traffic network operations, in general, and traffic congestion in particular, due to the limited amount of research conducted on those topics. This is primarily attributed to the lack of readily available ride-pooling trip data and the limited ability of commercial simulation software to simulate TNC trips, including ride-pooling trips.

In our previous work [6–9], we addressed some of these limitations by (a) collecting TNC trip data from a survey of Uber/Lyft drivers in the Birmingham, AL region, and (b) showcasing the feasibility of modeling TNC services using the MATSim (Multi-Agent Transport Simulation) platform and the Birmingham, AL transportation network. To address the limitation of acquiring TNC trip data, the survey of Uber/Lyft drivers in the Birmingham metro area acted as a seed for generating population plans in the study area using the synthetic population technique. Uber trips were incorporated into the day plans of the Birmingham MATSim model using MATSim's Taxi extension (org.matsim.contrib.taxi). More details on this effort are available in [6]. It should be noted that our earlier examination of the impact of ride-hailing TNC services on traffic operations was limited to individual ride requests, since ride-pooling services were not available in the Birmingham region [7]. However, it is of interest to understand how availability of various types of ride-pooling TNC services can affect operations and the likely impacts that such services can have on traffic congestion in the service area.

In earlier studies, researchers [10–12] analyzed two distinct types of DRT transportation services, namely door-to-door (d2d) and stop-based (sB) [10–12]. Door-to-door service involves picking up and dropping off passengers at their preferred location, thus providing a highly convenient transportation service similar to a taxi. In contrast, a stop-based service operates by providing transportation to/from specific stops where passengers can meet and board the vehicle. These stops are typically located at frequently used destinations such as bus stops. A stop-based service may be more affordable for riders, compared to door-to-door. It is, however, necessary that passengers get to the location where the stop is located in order to meet their ride, which can be challenging and inconvenient for many riders, especially for those with mobility impairments.

This paper reports on a study that investigated and documented operational performance impacts of door-to-door and stop-based ride-pooling services operating in the Birmingham region. Building on our earlier work, we expanded the Birmingham MATSim model to introduce ride-pooling TNC services using the MATSim's Demand Responsive Transit (DRT) module. This allowed us to simulate various fleet sizes of TNCs in order to quantify the impact of ride-pooling services on urban congestion. The updated Birmingham MATSim simulation model provided an excellent test bed for running experiments as it was capable of simulating trips that involved personal vehicles, ride-pooling TNC trips, public transit trips, and walking trips, all in the same network.

## 2. Literature Review

As ride-pooling services became more prevalent over the last few years, some researchers attempted to study their impacts on transportation system operations. The results are mixed as some studies suggest congestion mitigation as a result of introduction of TNC ride-pooling services, whereas others suggest that such services lead to exacerbation of urban congestion. For example, a study conducted by Chen et al. [13] used ride-sourcing provider data and online surveys to examine ride-pooling impacts in Hangzhoo and sug-

gested that such services result in a reduction of VKT by 58,124 VKT per day. Zhu and Mo [14] found that ride-pooling with a buffer time of 60 s led to an aggregate reduction of VKT of 8.21% in Haikou, China, compared with the traditional ride-hailing operation. Bischoff et al. [15] used a dynamic simulation approach to demonstrate that the overall number of VKTs may be reduced by between 15 and 20% in Berlin, if taxi rides are shared. Tirachini and Gomez-Lobo [16] conducted a Monte Carlo simulation study in Santiago, Chile, and concluded that while ride-hailing applications are expected to increase VKT, shared or pooled ride-hailing has the potential to decrease VKT. The authors point to the average occupancy rate of ride-hailing trips as a key parameter for VKT and suggest that an average occupancy rate of 2.9 passengers or more is needed to materialize congestion benefits.

On the other hand, some scholars argue that ride-pooling services might increase traffic congestion. For example, in their study of Demand Responsive Transit (DRT), Kagho et al. [17] found that the introduction of such a service was likely to increase overall VKT slightly in Wayne County, Michigan. Another study conducted in the same area [18] confirmed this finding and concluded that the introduction of DRT can increase the VKT by 22%. Furthermore, a study by Wu and MacKenzie [19] used 2017 US National Household Travel Survey data to examine the heterogeneous VKT effects on ride-sharing across population groups and reported an estimated net increase of 12.55 million VKT per day in the US due to ridesharing, compared to a case where all NHTS 2017 respondents are considered to be non-users of ride-sharing services. The authors further suggested that the impact of ride-hailing services on transportation network operational performance will continue to change dynamically in the future, as TNC services themselves and users' adoption practices continue to evolve over time.

Some other studies suggest that the impacts of ride-hailing applications on VKT and congestion are still inconclusive, including a study in Vancouver, Canada [20], and another one by [21] in San Francisco. Clewlow and Mishra [22] also recognize that VKT changes are unknown, and detailed information on the number of trips that ride-hailing applications are attracting from other modes (number, type, distance traveled, etc.) is needed to quantify such impacts. As Abouelela et al. [23] state, the reduction of VKT using ride-pooling services is possible, but it depends on a number of factors such as the use of suitable vehicle sizes to accommodate pooling service occupancy needs, the type of replaced modes (e.g., automobile versus walking or transit), and modes used to access and egress from the service [16,23].

Some studies investigated the impact of ride-pooling services on average travel time. For example, Li et al. [24]; Chau et al. [25]; and Fielbaum and Alonso-Mora [26] calculated the differences in travel time and detours between individual ride-hailing and ride-pooled rides, while Schwieterman and Smith [27] compared average trip times between Uber Pool and transit trips in Chicago. Leich and Bischoff [28] found that the average total travel time spent by the passengers from origin to destination was reduced by less than 2 min (3.5%) in simulated scenarios of door-to-door demand responsive services in Berlin, Germany. A summary of additional studies on ride-pooling impacts on VKT and aggregate travel time is available in [14].

To date, most studies on the impact of ride-pooling services have been conducted in cities/counties larger in population size than Birmingham metro area, which has a population of 0.89 million [29]. For instance, studies have been carried out in: Hangzhou, China [13] (8.24 million population [30]); Haikou City, China [14] (2.02 million population [30]); Berlin, Germany [15] (3.57 million population [30]); Santiago, Chile [16] (6.9 million population [30]); and Wayne County, Michigan [17,18] (1.77 million population [31]). Since the population size affects transportation demand, further research is needed to investigate how ride-pooling services affect the operational efficiency of the transportation system in medium-sized cities (population between 350,000–999,999).

The purpose of this study is to address this need by examining and documenting the impact of ride-pooling services in Birmingham, AL, a medium-sized city in the Southeast.

Comparisons of measures of traffic performance (e.g., Vehicle Kilometers Traveled, travel time, and user waiting time) in the presence of door-to-door and stop-based ride-pooling services provide valuable insights on their impacts and help identify conditions under which such services yield the greatest benefits. Moreover, the study considers various fleet sizes and provides guidance on selection of a proper fleet size in order to balance the needs of the riders, drivers, and traffic network operators.

## 3. Methodology

The aim of this study was to quantify the operational performance impacts of ride-pooling services (e.g., Uber Pool and Lyft Line) in the Birmingham metro transportation network. As of the time of the study, no ride-pooling services were offered by Uber and Lyft in Birmingham. Thus, in the absence of field data, we simulated the Birmingham transportation network (a) under baseline conditions and (b) assuming the presence of ride-pooling services. The simulation outputs allowed us to obtain and compare selected performance measures, including trips by mode, VKT, detour distances, mean passenger wait- and in-vehicle travel time, among others.

### 3.1. Study Area

The study area is located in north central Alabama and covers Jefferson and Shelby counties. Jefferson County encompasses an area of 1119 square miles [32], while Shelby County covers 800 square miles [33]. The population density in Jefferson County is 592 people per square mile [34], whereas in Shelby County it is 274 individuals per square mile [35]. The Birmingham–Jefferson County Transit Authority (BJCTA) operates a public transportation system, which includes buses and a paratransit service. Around 95% of Birmingham residents commute to work by driving or carpooling [36]. This corresponds to the findings of a commuter survey carried out in the Greater Birmingham region, which revealed that over 90% of transportation users use private automobiles to commute [37]. According to [36], the average individual driving distance per day within the Greater Birmingham area is approximately 34.1 miles.

### 3.2. Simulation Model Selection

The simulation platform used in this study was MATSim. MATSim is an open-source agent-based and activity-based microsimulation framework that is implemented as a Java application and is capable of simulating large-scale scenarios for various transportation options [38]. The model uses daily travel plans of all transportation users (population) and executes them on the road network to simulate traffic. Using a scoring mechanism and through a re-planning process, agents (transportation users) seek possibilities to optimize their plan at each iteration. The iterative process continues as long as the overall score of the population continues to increase. A detailed description of MATSim features can be found in Horni et al. [39,40].

The model selection in this study was based on MATSim's effectiveness in simulating TNC operations as demonstrated in the literature and confirmed by our earlier research efforts [7,8,38]. In particular, MATSim's Demand Responsive Transport (DRT) extension is a key software feature that gave MATSim a unique advantage over other available transportation software options considered. This extension enables the simulation of on-demand ride-pooling services [11,15,41], and makes MATSim an excellent choice for a simulation platform for meeting the objectives of our study.

### 3.3. Simulation Study Experimental Design

We developed the ride-pooling scenarios based on consideration of two attributes: ride-pooling service type and fleet size. Specifically, we considered two types of ride-pooling services, namely door-to-door (d2d) and stop-based (sB), and three fleet sizes (i.e., 200, 400, and 800 vehicles). The maximum acceptable waiting time was set to 5 min, as in other literature studies [15,41]. By combining the different options, 6 ride-pooling scenarios

were developed for further analysis. In addition, we considered the baseline scenario where we simulated no ride-pooling operations [7] and used it to facilitate comparisons.

### 3.4. Birmingham MATSim Simulation Model

Every MATSim model is built around a configuration file, a network file, and a population file. The configuration file defines the parameters and settings of the model that determine how the model behaves and provides access to the settings at runtime. The network file describes the details of the nodes and links that compose the transportation network and associated attributes (e.g., node coordinates, link length, number of lanes, speed, and capacity). The population file provides information about travel demand, i.e., lists of agents (travelers) and their day plans (trips). The population file contains the list of agents. Each agent has a list of plans, and each plan contains a list of activities and legs that describe each agent's planned actions.

Building on our earlier work [6,7], we adopted the Birmingham MATSim simulation model and made necessary modifications to the model to meet the needs of the current study. More specifically, in our previous research, we utilized MATSim's Taxi extension to simulate individual ride requests using the Birmingham MATSim model. In the current study, we adopted MATSim's Demand Responsive Transport (DRT) extension to simulate on-demand ride-pooling services in the Birmingham network. This extension provides the ability to fit multiple trip requests within a single TNC vehicle, a critical requirement for meeting the objectives of our study.

MATSim input files for the Birmingham model (e.g., network file, plans file, vehicles file, transit schedule file, and transit vehicles file) were adopted from [6] and allowed for multi-modal simulation of traffic operations in the Birmingham network (including generation of trips by passenger car, transit bus, and walking). We also added the stops file for the stop-based (sB) scenario. The stops file was created based on existing bus stops in the study area and included a total of 1856 bus stops which were geocoded as an XML input file.

The DRT MATSim extension was used to simulate ride-pooling trips for the study door-to-door (d2d) and stop-based (sB) scenarios. The DRT module allows the simulation of pooled rides in MATSim with one or several DRT operators, each of them having its own characteristics, such as the vehicle fleet, detour, or scoring parameters. For the purposes of maintaining a realistic comparison with the baseline scenario in [7], the configuration file in our study used identical parameters with those in our earlier work [6,7]. The DRT module configuration file, created specifically for this study, defined the DRT operational characteristics and selected reasonable parameter threshold values based on recommendations from earlier studies and local considerations. For example, studies conducted in Germany [10,11,41,42] used a 30 s stop time duration for pickup and drop off of passengers, Bischoff et al. [43] assumed a stop duration of 60 s, and a study conducted by [17] in Wayne County, Michigan, set the stop duration to 105 s. In our study, we set a stop time duration of 60 s for passengers being picked up and dropped off at each stop. The maximum number of passengers per vehicle was set to 4 and the maximum detour time in our study was 8 min.

In order to improve the computational efficiency of the simulation and in accordance with earlier studies that used MATSim to simulate city-level transportation networks, 10% of the total population in Birmingham was used as input [6,9,40,44]. Thus, plans were executed using a population size of 69,826. MATSim generates output data that can be used to analyze results, as well as to monitor the simulation setup progress. A link stats file containing hourly count values and travel times on every network link is produced in each iteration, and network wide measures of effectiveness (MOEs) can be obtained. In our study, we evaluated the Birmingham network performance under the six study ride-pooling scenarios using vehicle kilometers traveled (VKT) and compared with baseline conditions (no TNC presence). We also considered mode shifts toward the ride-pooling service and their impacts on network operations. The results are summarized next.

## 4. Results

### 4.1. Status of Ride-Pooling Vehicles

Figure 1 depicts the number and status of ride-pooling d2d and sB service vehicles by hour of day (24-h total) assuming different fleet sizes (200, 400, and 800 ride-pooling vehicles). In Figure 1 the status of ride-pooling vehicles, i.e., en-route (requested but not completed), departing, and arriving, during each hour of the day is clearly marked by green, red, and blue lines, respectively. We can observe that the number of ride-pooling vehicles en-route to pick up customers increases as the number of TNC vehicles increases (from 200 to 400 and 800), and peaks during the a.m. and p.m. peak traffic periods. According to Figure 1, there are more vehicles en-route for d2d compared to those in the sB scenario when 800 vehicles are added to the network. However, this reversal in the trend can be explained by the total number of ride-pooling trips, which can be seen in Table 1. When 800 TNC vehicles were added to the network, the d2d service had a higher number of ride-pooling trips than sB. In contrast, sB trips increased by approximately 7.6%, which indicates that the demand for sB service is nearing the saturation point.

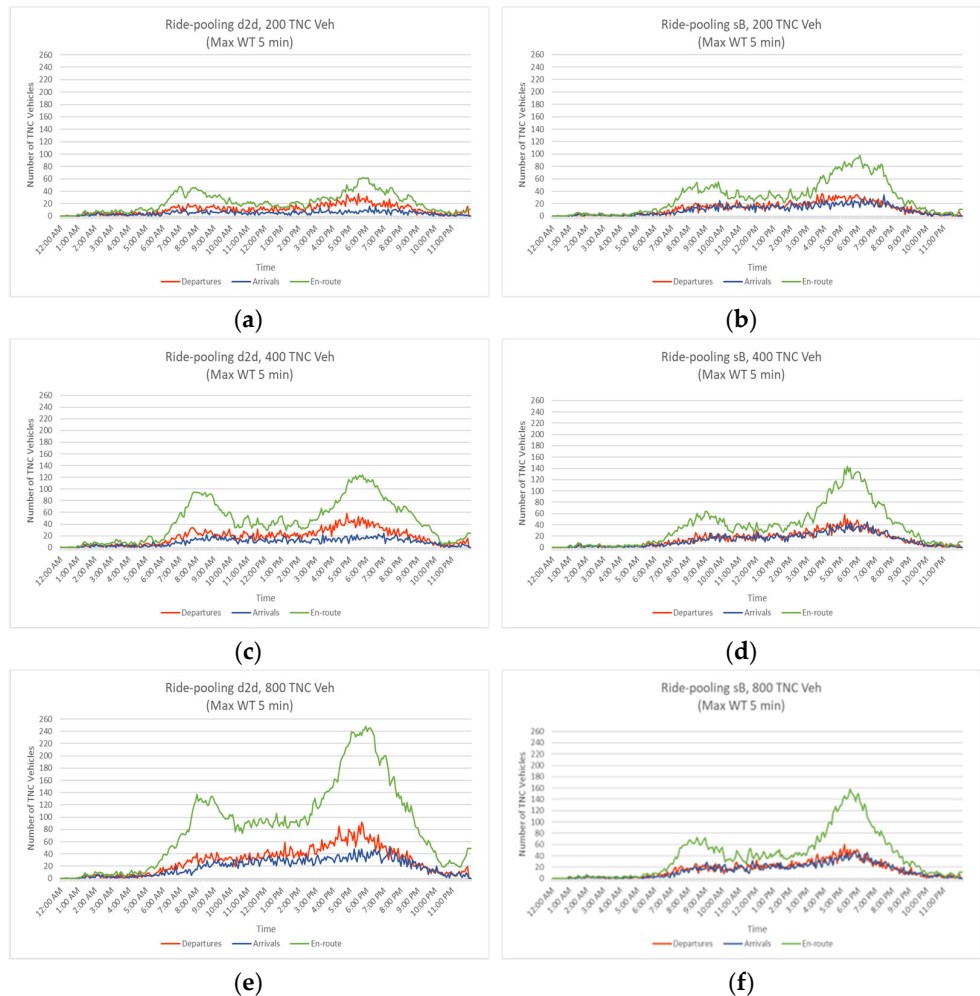

**Figure 1.** Ride-pooling Vehicle Status by Hour of Day for Various Fleet Sizes. (**a**) Ride-pooling (d2d) 200 TNC Veh; (**b**) ride-pooling (sB) 200 TNC Veh; (**c**) ride-pooling (d2d) 400 TNC Veh; (**d**) ride-pooling (sB) 400 TNC Veh; (**e**) ride-pooling (d2d) 800 TNC Veh; (**f**) ride-pooling (sB) 800 TNC Veh.

### 4.2. Vehicle Occupancy Profiles

Figure 2 illustrates vehicle occupancy profiles for d2d and sB ride-pooling services under various fleet sizes considered. Color codes indicate whether a TNC vehicle is a stay (gray color), carries zero passengers (purple), carries one passenger (yellow), or

accommodates two passengers (green), three passengers (blue), or four passengers (red) as a ride-pool. During the stay period, a vehicle is either parked or idle and waiting for the next trip request to be made. According to Zwick and Axhausen [45], if the status of the TNC vehicle is zero passengers, the TNC vehicle is either rebalancing or is on its way to pick up a rider.

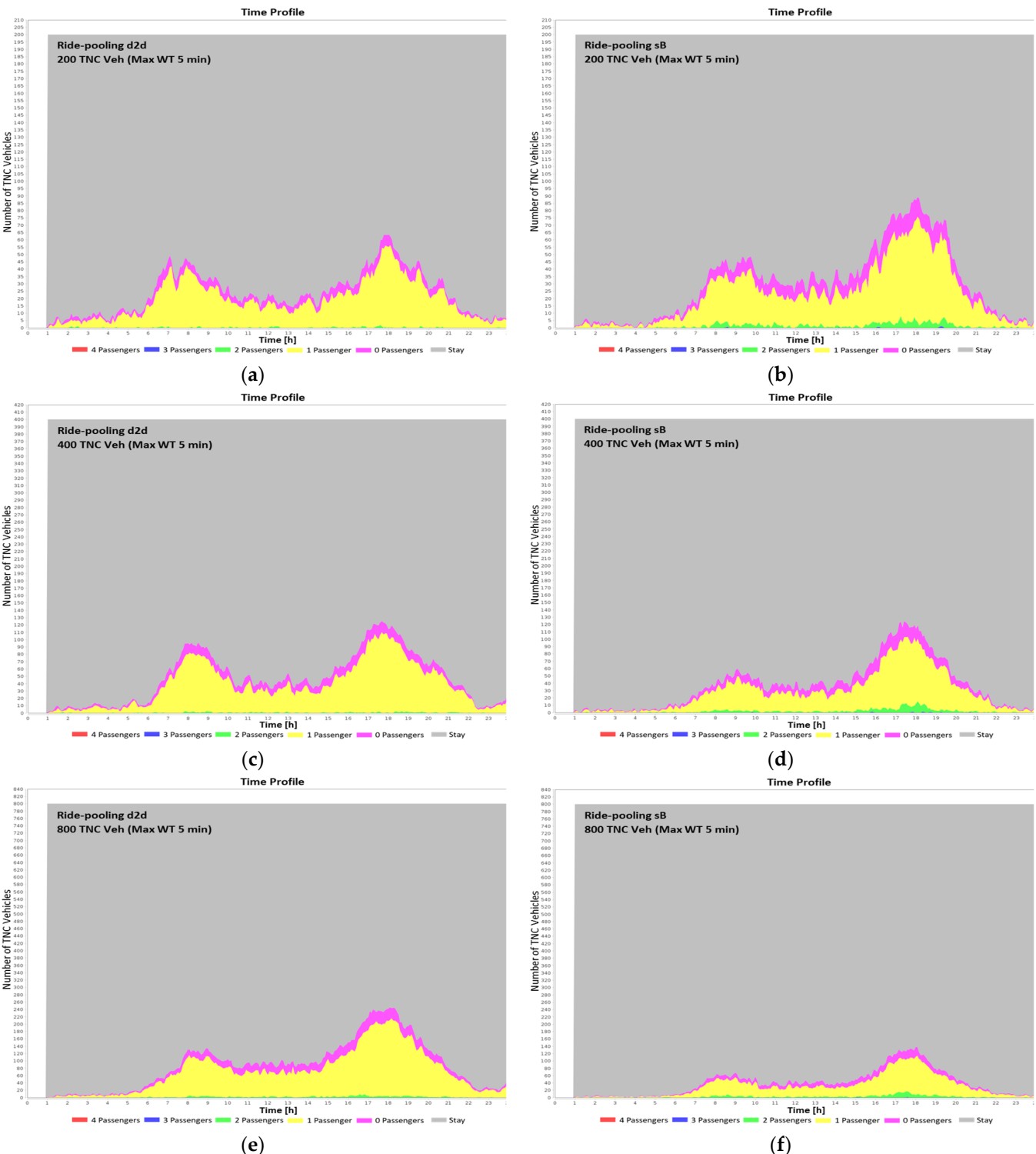

**Figure 2.** Vehicle Occupancy Profiles for Various Fleet Sizes. (**a**) Ride-pooling (d2d) 200 TNC Veh; (**b**) ride-pooling (sB) 200 TNC Veh; (**c**) ride-pooling (d2d) 400 TNC Veh; (**d**) ride-pooling (sB) 400 TNC Veh; (**e**) ride-pooling (d2d) 800 TNC Veh; (**f**) ride-pooling (sB) 800 TNC Veh.

In all study scenarios, we can observe that ride-pooling vehicles at stay (gray color) are overrepresented, meaning that the demand for ride-pooling services in the Birmingham region is below the available supply for service. This is the case even during the a.m. and p.m. peak periods. The results show that even a fleet of 200 ride-pooling vehicles can comfortably meet the current demand for service in the Birmingham region at all times. Figure 2 also shows that the vast majority of passenger-carrying TNC vehicles are single-passenger rides (yellow) and neither the d2d service nor the sB service serve any trip requests from more than three passengers in a single ride. Thus, vehicles that can accommodate three passengers meet the service needs of the Birmingham area and can be used as ride-pooling vehicles. Although Figure 2 indicates that most car-pooling trips in our study were single-passenger trips, it can still be concluded that the introduction of ride-pooling has a positive impact on the traffic network operation. This is due to two main reasons: (a) an observed mode shift from automobile use to transit, walking, and ride-pooling, and (b) a reduction in the total number of vehicular trips (combining both private auto and ride-pooling) for both d2d and sB ride-pooling services, as demonstrated in Table 1.

### 4.3. Impact of Ride-Pooling Service Availability on Modal Choice

Table 1 summarizes the distribution of trips by mode for the baseline and the six ride-pooling service scenarios analyzed in this study, under baseline conditions, at total of 144,014 vehicle trips were performed (all by a private automobile). However, when TNCs were introduced, the total number of vehicle trips (private automobile and ride-pooling combined) decreased in all ride-pooling scenarios considered. This represents a decrease in the range of 5.45% to 2.81% compared to the baseline (or 136,163 to 139,974 compared to 144,014 vehicle trips). The results in Table 1 also show that as the TNC fleet size increases, the number of ride-pooling trips increases as well. It is worth noting that the changes are more pronounced under d2d scenarios versus sB scenarios. Specifically, as the TNC fleet size changes from 200 to 400 and 800 vehicles, the number of ride-pooling trips almost doubles under d2d scenarios, while the number of ride-pooling trips under sB scenarios increases at a lower pace. Ride-pooling scenarios also show more transit/walk trips than those of the baseline scenario, however, increase in ride-pooling fleet sizes reduces the shift toward transit/walking modes.

**Table 1.** Statistics of Executed Plans-Trips by Mode (Max Acceptable Wait Time: 5 min).

| TNC Fleet Size (Vehicles) | Scenario | Transit Trips (Total Ridership) | Walk Trips | Private Auto Trips | Ride-Pooling Trips | Vehicle Trips (Private Auto + Ride-Pooling) | Change in Vehicle Trips Due to Ride-Pooling (Baseline—Number of Vehicle Trips) | % Change in Vehicle Trips Due to Ride-Pooling Compared to Baseline |
|---|---|---|---|---|---|---|---|---|
| 0 TNCs | Baseline | 2648 | 5172 | 144,014 | 0 | 144,014 | 0 | 0% |
| 200 TNCs | d2d | 4590 | 8115 | 135,156 | 1386 | 136,542 | −7472 | −5.19% |
| | sB | 3649 | 7987 | 136,167 | 2909 | 139,076 | −4938 | −3.43% |
| 400 TNCs | d2d | 4359 | 7693 | 133,445 | 2718 | 136,163 | −7851 | −5.45% |
| | sB | 3423 | 7840 | 135,914 | 3853 | 139,767 | −4247 | −2.95% |
| 800 TNCs | d2d | 3919 | 7093 | 131,333 | 5420 | 136,753 | −7261 | −5.04% |
| | sB | 3272 | 7851 | 135,828 | 4146 | 139,974 | −4040 | −2.81% |

*4.4. Impact of Ride-Pooling Service Availability on Network-Wide Operations*

4.4.1. Total Daily Network VKT

Using Equation (1) below, we calculated the total daily VKT in the Birmingham network for each of the six ride-pooling scenarios in our study. The results are summarized in Table 2.

$$VKT_{Day} = \sum_{h=0}^{h=23} Vehicle\ Count\ /\ Hour \times \frac{Link\ Length\ (\text{m})}{1000\ \frac{\text{m}}{\text{km}}} \tag{1}$$

**Table 2.** Total Daily VKT under Various Scenarios (Max Acceptable Wait Time: 5 min).

| TNC Fleet Size (Vehicles) | Scenario | Total Daily VKT | Change in Total Daily VKT (Baseline—Ride-Pooling Scenario) | VKT % Diff. to Baseline |
|---|---|---|---|---|
| 0 TNCs | Baseline | 2,265,716 | | |
| 200 TNCs | d2d | 2,134,646 | −131,070 | −5.78% |
| | sB | 2,204,292 | −61,424 | −2.71% |
| 400 TNCs | d2d | 2,157,837 | −107,879 | −4.76% |
| | sB | 2,209,273 | −56,443 | −2.49% |
| 800 TNCs | d2d | 2,193,750 | −71,966 | −3.18% |
| | sB | 2,212,335 | −53,381 | −2.36% |

It can be observed that ride-pooling services scenarios show a reduction in the total VKT, when compared to the baseline scenario. As shown in Table 2, ride-pooling d2d scenarios result in a reduction in total daily VKT of up to 5.78% (or up to 131,070 fewer VKT) in comparison to the baseline scenario, while sB scenarios show more moderate improvements (up to 2.71% or 61,424 VKT reduction over the baseline). We can also see that under all ride-pooling scenarios, a TNC fleet size of 200 vehicles yields the best VKT results and larger fleet sizes result in an increase in the total daily VKT.

Further analysis indicates that the total hourly VKT in the presence of ride-pooling services peaked during the a.m. and p.m. traffic peak periods (7 to 8 a.m. and 4 to 5 p.m.), however, it remained below that of the baseline as illustrated in Figure 3. The afternoon peak hours experienced the highest total hourly VKT values reported throughout the day. It is worth noting that the difference in VKT among scenarios tends to be relatively small, but, overall, smaller fleet sizes result in lower total hourly VKT, an observation that is consistent with the total VKT results reported earlier.

4.4.2. Ride-Pooling Daily VKT

Figure 4 illustrates the daily VKT for ride-pooling trips in the Birmingham region for two ride-pooling service types (d2d and sB) and for three fleet sizes (200, 400, 800 available ride-pooling vehicles). As shown in Figure 4, an increase in the size of the TNC fleet results in an increase in the ride-pooling total daily VKT for both the d2d and sB scenarios. It is also observed that the d2d service generates a greater number of VKT than the sB, as the fleet size increases from 200 to 400 and 800 vehicles. This can be attributed to the longer mean travel distance associated with d2d compared to sB. Additionally, sB service vehicles only pick up and drop off passengers at designated locations within the study area, thus covering a shorter range of service. However, d2d service vehicles can pick up and drop off passengers from anywhere within the network, and cover service requests from across the entire service area. The results confirm that the sB ride-pooling service is more desirable from the operators' perspective, as it results in lower VKT, even though it is often less desirable from the perspective of the user that typically favors d2d services due to the added convenience.

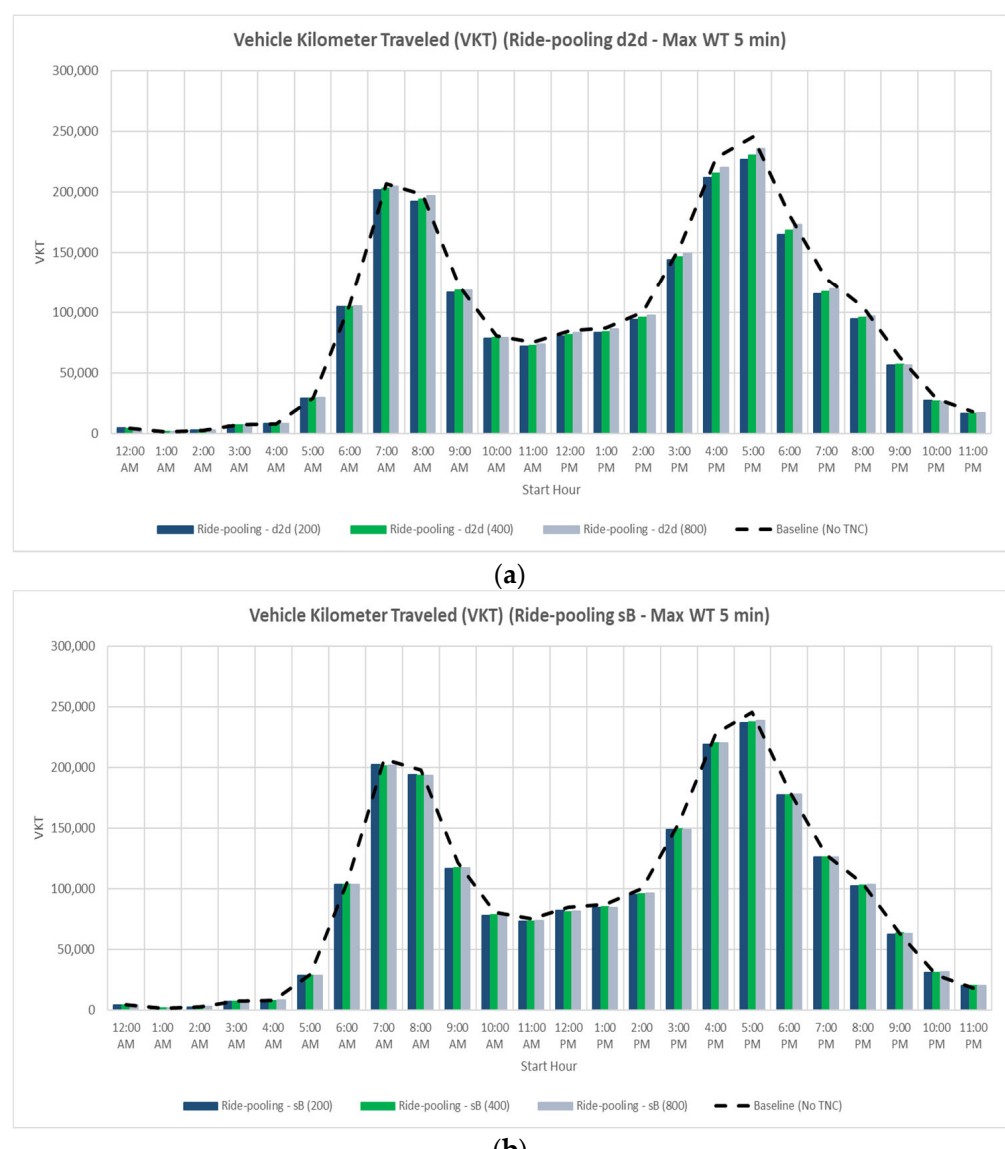

**Figure 3.** Hourly Distribution of Total VKT under Various Scenarios. (**a**) ride-pooling (d2d); (**b**) ride-pooling (sB).

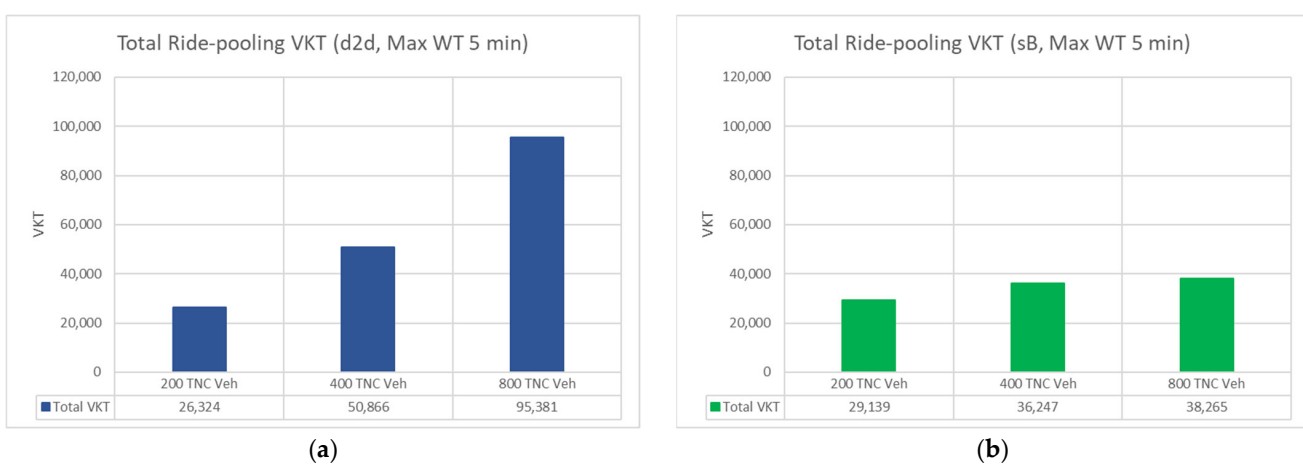

**Figure 4.** Total VKT for Ride-pooling Trips under Various Scenarios. (**a**) ride-pooling (d2d); (**b**) ride-pooling (sB).

### 4.4.3. Ride-Pooling Vehicle Distance Travelled

Table 3 reports daily distance traveled by ride-pooling vehicles in the Birmingham network under the assumption of max acceptable wait time of 5 min for two types of operations (d2d and sB service), and three fleet sizes (200, 400, and 800 vehicles). The daily distance that the ride-pooling vehicles cover while empty is also reported and used to calculate the empty ratio. Empty ratio refers to the ratio of travel distance covered by ride-pooling vehicles with no passenger on board over the total travel distance covered by TNC vehicles.

**Table 3.** Daily Distance Traveled by Ride-pooling Vehicles (Max Acceptable Wait Time: 5 min).

| TNC Fleet Size (Vehicles) | Scenario | Total Daily Distance Traveled (km) | Daily Distance Traveled While Empty (km) | Empty Ratio (%) | Total Detour Distance (km) |
|---|---|---|---|---|---|
| 200 TNC Veh | d2d | 26,324 | 2145 | 8.15% | 1395 |
|  | sB | 29,139 | 3248 | 11.15% | 2974 |
| 400 TNC Veh | d2d | 50,866 | 4132 | 8.12% | 2743 |
|  | sB | 36,247 | 3518 | 9.71% | 3948 |
| 800 TNC Veh | d2d | 95,381 | 7354 | 7.71% | 5492 |
|  | sB | 38,265 | 3455 | 9.03% | 4250 |

From Table 3, it is evident that the daily distance traveled while empty increased as the TNC fleet size increased in size. Additionally, there was a corresponding decrease in the empty ratio as the size of the TNC fleet increased. This trend can be attributed to mode shifts that resulted in an increase in ride-pooling demand, which occurs with an increase in TNC fleet size. Table 1 indicates that the number of TNC trip demands increased as the TNC fleet size increased, while the ride request rejection rate decreased as shown in Table 4 below. These findings align with earlier research, which demonstrated a decrease in the proportion of empty vehicles on the road when the DRT fleet size increased, resulting in more vehicles distributed in the network [17]. Furthermore, the empty ratios were higher under the assumption of sB operations, compared to d2d service. The sB service also resulted in higher total detour distance, compared to the d2d service for fleet sizes of 200 and 400 TNC vehicles.

**Table 4.** Ride-pooling Customer Service Time and Ride Request Rejection Rate for Various Scenarios (Max Acceptable Wait Time: 5 min).

| TNC Fleet Size (Vehicles) | Scenario | Mean Passenger Wait Time (s) | Mean In-Vehicle Travel Time (IVTT) (s) | Mean Passenger Service Time (s) | Mean Travel Distance (m) | Ride Request Rejection Rate (%) |
|---|---|---|---|---|---|---|
| 200 TNC Veh | d2d | 219 | 1153 | 1372 | 17,486 | 55% |
|  | sB | 182 | 703 | 885 | 9415 | 18% |
| 400 TNC Veh | d2d | 220 | 1195 | 1415 | 17,271 | 47% |
|  | sB | 164 | 669 | 833 | 9050 | 10% |
| 800 TNC Veh | d2d | 214 | 1132 | 1346 | 16,485 | 33% |
|  | sB | 163 | 676 | 839 | 9011 | 8% |

### 4.4.4. Customer Service Time and Ride Request Rejection Rate

Table 4 summarizes simulated average customer wait times, in-vehicle travel times, mean passenger service times, and ride request rejection rates for all study scenarios. Mean passenger service times are calculated as the sum of average customer wait times, and average in-vehicle travel times. The results show that the d2d ride-pooling service led to higher average customer wait times and higher in-vehicle travel times, compared to the sB service. As a result, the total customer service time for the d2d service was consistently greater than that of the sB service when accounting for fleet size. According to the results in

Table 4, the average passenger wait time for ride-pooling d2d scenarios was up to 3.7 min (3 min for ride-pooling sB scenarios).

According to Table 4, as the TNC fleet size increases for both d2d and sB ride-pooling services, the mean travel distance decreases. This phenomenon can be attributed to the increased availability of TNC vehicles, making it more convenient to match passengers with nearby drivers. This, in turn, could encourage individuals to utilize the TNC service more frequently, particularly for shorter distances.

The ride request rejection rate was determined by dividing the number of rejected trips by the total requested number of pooled rides. The ride request rejection rate for d2d ride-pooling scenarios reached up to 55% for TNC fleet size of 200 vehicles, while the rejection rate for sB ride-pooling scenarios was considerably lower (up to 18%). The results in Table 4 further confirm that an increase in the TNC fleet size results in a reduction in ride request rejection rates. For example, when the TNC fleet size increases from 200 to 800 vehicles, the rejection rate falls from 55% to 33% for d2d ride-pooling services, and from 18% to 8% for sB ride-pooling services.

## 5. Summary and Conclusions

This study aimed at assessing how ride-pooling services affect traffic operations throughout the Birmingham, AL metro area. Using the MATSim simulation platform, a baseline scenario (no TNC vehicles) was simulated for the Birmingham network [6,7], along with six ride-pooling scenarios that represented two variations in the ride-pooling service type (d2d and sB) and three TNC fleet sizes (200, 400, and 800 vehicles). The ride-pooling scenarios were simulated using MATSim's DRT extension and assumed a maximum acceptable waiting time of 5 min. Trips for transit, walk, private automobile, and ride-pooling travel modes were obtained from each simulated scenario and compared to the baseline in order to determine modal shifts in the presence of TNC ride-pooling services. The VKT for each scenario was obtained from the DRT output of MATSim and used to quantify the impacts of the ride-pooling service on network operations across the Birmingham network over a period of 24 h. In addition, consideration of the TNC vehicle status over a 24-h period for each study provided useful information to the TNC operator about the operational efficiency of ride-pooling options for various TNC fleet sizes. Last, but not least, ride-pooling customer service time and ride request rejection rates were evaluated to gain insights on the quality of customer service offered by ride-pooling services under various study scenarios.

Our results showed that the introduction of ride-pooling TNC services in the Birmingham region can be beneficial from a network operation perspective, as it has the potential to reduce VKT by up to 5.78% for d2d services, and up to 2.71% for sB services, compared to the baseline scenario (no TNC service). These results are consistent with findings from studies involving larger cities [13–16] that reported that an 8–20% reduction in VKT is possible in the presence of ride-pooling services.

When comparing the simulated VKT from this study to our earlier research findings in [7], we can clearly see the effectiveness of ride-pooling services. For the same baseline traffic network, similar demand conditions, and for a fleet of 200 TNC vehicles, ride-pooling d2d TNC services reduce VKT by 5.78% in the Birmingham network, whereas individual ride TNC requests increase VKT by 22% [7], compared to the baseline scenario. Similar conclusions can be drawn from the comparison of different fleet sizes and the comparison between sB and individual ride TNC services, and are in agreement with findings reported by Tirachini and Gomez-Lobo [16] from a simulation study in Santiago, Chile. Thus, it is recommended that TNC companies operating in the Birmingham region consider expanding their services to provide car-pooling services, in addition to individual TNC rides, given the anticipating benefits to the transportation network performance from the operation of ride-pooling TNC services.

The study concludes that an increase in the TNC fleet size leads to a decrease in the vehicle-empty ratio for both d2d and sB ride-pooling services. However, the total daily

distance traveled by ride-pooling vehicles increases with an increase in the TNC fleet size for both d2d and sB services. The most significant increase in the total daily stance traveled occurred for d2d, with an increase of up to 93.2% (from 26,324 km to 50,866 km) when the fleet size increased from 200 to 400 TNC vehicles. Additionally, the study found an increase of up to 24.4% (from 29,136 km to 36,247 km) for sB when the fleet size increased from 200 to 400 TNC vehicles. The observed trend can be explained by the shift in transportation modes from automobile trips to ride-pooling services. This shift results in an increase in demand for ride-pooling services, which is seen with an increase in the TNC fleet size.

The study findings also show that customer service time is almost insensitive to changes in the size of the TNC fleet for both ride-pooling services considered (i.e., d2d and sB). This observation is consistent with earlier studies, including [43], which also found that an increase in the TNC fleet size had a minimal or no effect on the in-vehicle travel time for both d2d and sB ride-pooling services. When comparing the two service options, our results suggest that the sB ride-pooling service was the most effective ride-pooling option, as it resulted in lower average customer service times when compared to the d2d service.

The study found that with an increase in the size of the TNC fleet, both d2d and sB services experienced a reduction in mean travel distance. The most significant decrease in the mean travel distance occurred for d2d, with a reduction of up to 4.55% (from 17,271 m to 16,485 m) when the fleet size increased from 400 to 800 TNC vehicles. Additionally, the study results showed a decrease of up to 3.88% (from 9415 m to 9050 m) for sB when the fleet size increased from 200 to 400 TNC vehicles.

The study findings further confirm that ride request rejection rates fall as the TNC fleet size increases. For instance, for sB service, the ride rejection rate fell from 18% to 10% (a 44% reduction) when the TNC fleet size increased from 200 to 400 vehicles. A further 20% reduction was observed when the fleet size increased from 400 to 800 vehicles. These findings are consistent with results from a previous study [18], which reported that increasing the DRT fleet size from 150 to 250 vehicles resulted in a 36% decrease in rejection rates and a 28% reduction in the rejection rate when the DRT fleet size was increased from 250 to 350 vehicles [18]. It should be noted that the Birmingham simulation study assumed a maximum acceptable wait time of 5 min. This resulted in high ride request rejection rates, especially when TNC fleet sizes were low. Higher acceptable wait times (e.g., 10 min) are expected to reduce the ride request rejection rates and are recommended for consideration in follow-up studies.

Overall, this study contributes to a better understanding of how ride-pooling services impact traffic congestion in medium-sized cities such as Birmingham, AL. Moreover, the findings from this study can guide TNC providers and transportation authorities in their efforts to enhance TNC operations in medium-sized cities with similar characteristics and better serve the needs of transportation users in these regions.

## 6. Study Limitations

In this study, we are addressing a critical limitation in the literature, i.e., examining impacts of ride-pooling services on traffic operations in a medium-sized city. However, given that travel behaviors and local conditions vary from city to city, the findings of the study are only generalizable to settings that are similar to the city of Birmingham. Additional studies are recommended in other medium-sized metro areas across the US in the future to further expand the knowledge and understanding of the relationship between TNC service availability and traffic congestion. Due to difficulties in obtaining TNC trip data directly from Uber and Lyft for the Birmingham region, the study relied on local Uber drivers to extract trip records from their logs, which may not be representative of the entire TNC population. Furthermore, the use of a synthetic population to generate travel plans for all travelers in the network is a limitation in MATSim, as a synthetic population may not accurately reflect the travel behaviors of transportation users, including TNC users in their entirety [6,7]. However, this study still provided proof of the feasibility of modeling TNC services on the same simulation platform with automobile, transit, and walking trips using

MATSim, thus confirming its superiority over other traffic simulation platforms toward modeling multimodal operations, including ride-pooling TNC services.

A significant amount of knowledge can be gained from the analysis of TNC data, such as user behavior and travel patterns. Thus, this study further recommends considering such data in the estimation of mode choices and mode shifts. One challenge, noted earlier, is the difficulty of obtaining empirical data from TNCs, as such data are closely guarded by TNC providers [46]. However, some recent initiatives demonstrate promise toward data sharing between TNC providers and public agencies. One notable example is that of the State of California that requires TNC providers to submit Annual Reports Data. Such reports are made publicly available by the California Public Utilities Commission at a dedicated TNC Data Portal [47] after they have been redacted to remove any identifiable information. More wide-spread sharing of data between TNC companies and researchers is encouraged in the future and is expected to lead to a better representation of TNC trips in transportation modeling, and a more in-depth understanding of mode choices and modal shifts in markets where TNCs operate. This, in turn, can benefit transportation agencies and TNC providers as well, and assist them to better serve the needs of the traveling public.

**Author Contributions:** Conceptualization, F.S. and V.P.S.; methodology, F.S. and V.P.S.; formal analysis, F.S.; data curation, F.S. and J.K.; writing—original draft preparation, F.S. and V.P.S.; writing—review and editing, D.Y., J.K. and W.Y.; funding acquisition, V.P.S. and D.Y.; project administration, V.P.S. All authors have read and agreed to the published version of the manuscript.

**Funding:** This research was funded by US DOT STRIDE UTC grant number [69A3551747104] and the Alabama Research and Development Enhancement Fund Program (ARDEF) grant number [1ARDEF2103].

**Institutional Review Board Statement:** Not applicable.

**Informed Consent Statement:** Not applicable.

**Data Availability Statement:** Not applicable.

**Conflicts of Interest:** The authors declare no conflict of interest.

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
