# Peer review of "Operational Impacts of On-Demand Ride-Pooling Service Options in Birmingham, AL"

_futuretransp, doi:10.3390/futuretransp3020030_

Round 1

Reviewer 1 Report

The authors built upon earlier work they conducted investigating the operational performance of ride-pooling services. The authors used the MATSim Demand Response Transit Module to analyze ride-pooling services to the medium size city, Birmingham, Alabama. The simulation considered trips for personal vehicles, ride-pooling TNC trips, public transit, and walking trips within the same network. With no TNCs as a baseline, the authors looked at what happens to vehicle kilometers traveled when two types of ride-pooling services are added to the Birmingham, AL network. Two types of service: door-to-door and stop-based service were added, and the authors conducted simulations varying the fleet sizes to 200, 400, and 800 vehicles. Results did point to a reduction in vehicle kilometers traveled when TNCs were introduced. Results showed that a fleet of 200 ride-pooling vehicles are adequate to meet the demand for service in the Birmingham region. The ride-pool vehicle fleet size of 200 may be optimal for medium sized cities.

The article is well-written, flows well, and is overall easy to read. Each section is well-developed.

The authors do a good job of synthesizing the literature explaining that results are mixed regarding the urban introduction of TNCs and suggest the need to look at the introduction of TNCs in medium sized cities. They point out that some scholars argue that ride-pooling services might increase traffic congestion while others argue that it will not. They indicate that studies suggest that the impacts of ride-hailing applications are still inconclusive.

It is interesting to apply the MATSim Demand Response Transit Module to analyze the ride-pooling services to a medium size city. Much attention has been given to ride-pooling in large urban areas so it is admirable to see the analysis applied to a medium-sized city. This information adds to the body of literature and serves as a good case study for others to consider.

Author Response

We appreciate your time reviewing our paper and thank you for your kind support of our work. 

Reviewer 2 Report

The study findings also confirm that ride request rejection rates decrease as the TNC's fleet size increases. For example, for the sB service, the trip rejection rate decreased from 18% to 10% (44% reduction) when the TNC fleet increased from 200 to 400 vehicles. A further 20% reduction in fleet size from 400 to 800 vehicles These findings are consistent with a previous study  which reported that increasing the DRT fleet size from 150 to 250 vehicles resulted in a 36% reduction in rejections. from 250 to 350 vehicles .

A simulation study in Birmingham found that the maximum acceptable waiting time is 5 minutes. This resulted in high ride request rejection rates, if we increased the waiting time, it is likely that the number of rejections would decrease.

The study provides a better understanding of how ride-hailing services affect traffic congestion in mid-sized cities such as Birmingham, AL. In addition, the findings of this study can help TNC providers and transportation authorities in their efforts to improve TNC performance in medium-sized cities with similar characteristics and better meet the needs of transportation users in these regions.

The presented article suggests how to implement improvements in transportation systems to reduce the number of vehicles in the network. Clearly illustrated research results are presented, both according to d2d and sB transportation models, and according to fleet size. Literature sources suitable for 38 building years were used.

Key notes

Name the problem more clearly and summarize the results obtained during the research more specifically.

Feel free to express an opinion on the pros and cons of using d2d and sB.

In what type of medium-sized cities would a ride-hailing system work best?

Reviewer 3 Report

This paper investigates the impacts of ride-pooling and for doing so it uses an open-source agent-based simulation framework, suitable for modeling on-demand services. The topic is interesing and up-to-date. Also, the manuscript is organized well and it in general it is well-written. Yet, I have some comments that I think they should be addressed for further improving the quality of the paper and making it suitable for publication.

1. The innovative part of your paper should be highlighted in the last paragraph of literature review. Is the population size the only difference between your study and those studies that have been already published?

2. Despite the fact that the study area is being mentioned as the main differentiation from previous studies, there is not a subsection that describes the study area (e.g. total area, density, PuT system, existing modal share, average trip distance).

3. The conclusion section is just summarizing the simulation results. Since some of the simulation results create an impression to me, I would expect to read a critical discussion around these results. More specifically, the simulation results that created an imprssion to me and I would like to see being discussed are the following:

a. d2d results in a greater VKT decrease comparing with sB. I would assume that in sB a part of the trip is being made on foot instead of motorized vehicles and therefore VKT would be less. Where do you attribute this finding?

b. From the literature review I understand that vehicles' occupancy rate is probably the most critical factor with regards to the impact of ride-pooling on the network. From your results it can be concluded that ride-pooling has positive impact in the network despite the fact that the vast majority of the trips are with a single passenger. Please discuss this finding.

c. Figure 1, for 200 vehicles, en-route has higher peaks in the sB scenario, while for 800 vehicles en-route has much higher peaks for the d2d scenario. Where do you attribute this result?

d. Table 3 shows that as fleet size increases, empty ratio decreases. Yet, in lines 349-351 it is mentioned that empty ratio increased as fleet size increased. The text should be corrected and a critical discussion is needed about why an increase in the fleet size decreases the empty radio (I would expect the opposite result). 

e. Table 4, I see that sB scenarios are much better comparing with the d2d in all cases. This is reasonable with regards to the rejection rate, but thinking of a real-life operation of such services I can't fully understand why sB is that better with regards to service time (especially considering that the vast majority of trips are for a single passenger).

4. It would be useful to add possible limitations of your study in the end of the paper.

Reviewer 4 Report

In this research, the authors aim to expand the Birmingham MATSim model to introduce ride-pooling TNC services and simulate various fleet sizes of TNCs in order to quantify the impact of ride-pooling services on urban congestion. Since no ride-pooling services were available in Birmingham at the time of the study, the researchers used a simulation approach to compare the performance of the transportation network under baseline conditions and with the introduction of ride-pooling services. They used the MATSim simulation platform and considered two ride-pooling service types (door-to-door and stop-based) and three fleet sizes (200, 400, and 800 vehicles) to develop six ride-pooling scenarios.

Some suggestions:

-          The novelty of the work, in particular the relationship with and differences from the authors’ previously published work [6-9] need to be clarified. Otherwise it could be considered as ‘inappropriate self-citations’.

-          The study is based on data collected from the Birmingham, AL region, which may not be representative of other cities or regions. Therefore, the results of the study may not be generalizable to other areas.

-          It is not clear to me how available TNC data has helped the simulation.

-          The study assumes that ride-pooling services can reduce traffic congestion, but it does not consider the possibility that these services may actually increase traffic congestion by inducing additional demand for transportation services.

-          The simulation approach relies on assumptions and simplifications that may not accurately reflect the real-world situation. For example, the simulation assumes that all travelers will adopt ride-pooling services, which may not be realistic.

-          The study focuses on the impact of ride-pooling services on traffic congestion, but it does not address the impact on public transit ridership. What does the MATSim results show?

-          Transportation companies have recently expanded their operation to micromobility, with relevant and comparable data, methodology, and effects. This topic should be reviewed, particularly the use of empirical data from operators. Future research borrowing these concepts should be discussed. 

Round 2

Reviewer 3 Report

I am happy to see that the authors addressed all of my comments. I think the quality of the paper has been improved and it is now suitable for being published.

Author Response

We would like to express our gratitude for the positive response and valuable recommendations, which have greatly contributed to the improvement of our paper.

Reviewer 4 Report

The authors have addressed most of my comments, except the last one.

I understand that empirical data from TNC operators were not available for this study, and much similar studies. However, what I meant is that, such data is often available from micromobility operators, which is often used to study similar topics such as mode choices and shifts. A brief review of these data and studies can help readers understand alternative methodologies that can be used to study the same research topic, should such empirical data become available. 
